# Creating IoT-Enriched Learner-Centered Environments in Sports Science Higher Education during the Pandemic

**Rocsana Bucea-Manea-Țoniș** [1,*] , **Luciela Vasile** [2], **Rareș Stănescu** [2,*] **and Alina Moanță** [2]

1  Faculty of Physical Education and Sport, Spiru Haret University, 030045 Bucharest, Romania
2  Faculty of Physical Education and Sport, National University of Physical Education and Sport, 060057 Bucharest, Romania; luciela.vasile@unefs.ro (L.V.); alina.moanta@unefs.ro (A.M.)
*  Correspondence: rocsanamanea.mk@spiruharet.ro (R.B.-M.-Ț.); dcre@unefs.ro (R.S.)

**Abstract:** In the pandemic context, creating IoT-enriched learner-centered environments was not only a tendency but a requirement for sustainable teaching and learning in universities with sports science programs for theoretical classes and practical activities. Our study aims to assess both the extent to which the sports science academic environment has been prepared for online teaching and the key features of dedicated e-learning teaching and training in sports science to provide the highest-quality educational services in pandemic conditions. An online survey was administered to academic staff in the field of sports science from two Romanian universities. The results of the study reveal that online teaching has been associated with major changes in terms of methods and methodology, but also with a new dynamic of external and internal factors regarding teachers and their relationship with students. At the same time, it depends on a solid specific infrastructure and IoT facilities (MOOCs, VR/AR, mobile devices). As a mirror of the student-centered approach, universities in the field of sports science have experienced the same concerns about the outcomes of the educational process. In this regard, universities can become sustainable if they positively integrate e-learning into their teaching system and consolidate their quality standards from an e-learning perspective.

**Keywords:** online education; emergency remote teaching; sports science; IoT; MOOCs; VR simulation

## 1. Introduction

In the pandemic context, most universities in the field of sports science were very concerned about interrupting practical activities and training lessons. Most of the theoretical courses were delivered in the online environment, but for practical activities physical training was compulsory to develop teaching and practical competencies. As a civil protection mechanism, the total or partial interruption of face-to-face activities required the adoption of solutions to organize an efficient training process that would retain the characteristics of the student-centered learning process [1,2]. The decision to stimulate digitization and use active student-centered learning methods was made at both governmental and institutional levels [3]. In this context, education was disrupted, and virtual learning through distance facilities was predominant during the pandemic.

The new conditions for carrying out teaching activities imposed an approach that has led to major changes in the educational process as regards both the teaching methods and challenges addressed to the training of teachers, who were forced to use the virtual environment for the teaching-learning-evaluation process [4–6]. Thus, we can say that many elements of the academic environment have been influenced: the forms of organizing the teaching process, the teaching-learning-evaluation process, the level of digital teacher training, the teacher-student interaction and support services, especially for disadvantaged and vulnerable students.

Major changes in the way of life and work, in general [5], and in the education system, in particular, have led many specialists in the field to ask questions about the quality criteria

that the online teaching system must meet in different fields of science [2]. Each institution had to design its development plan to meet the needs of students and faculty [6] according to the characteristics of study programs and the concrete conditions of implementation (training level of human resources, material facilities, technology).

Although the conditions for conducting the teaching process are completely different, ranging from the physical environment to the virtual environment, the ultimate goals of the study programs have remained the same. On study completion, students earn their degrees attesting to the same skills as if they had attended face-to-face activities. All these concerns arose from the need to maintain the quality of education from a UNESCO perspective in terms of sustainable development goals to meet national quality standards in higher education institutions (HEIs) and remain credible in the professional development market and the labor market so that the degrees awarded reflect the scope of real professional competencies [7]. The effort was to ensure the sustainability of graduates' skills acquired through face-to-face curricula compared to those resulting from online and hybrid education.

Under these conditions, several questions arise:

— To what extent has the sports science academic environment been prepared for online teaching?
— What are the factors that condition the quality of educational services provided through the online system in sports science higher education?
— What characteristics of sports science HEIs can influence the quality of the e-learning process?
— What are the relationships between online teaching methods and students' transversal skills?

The paper aims to answer these questions by analyzing the opinions of teachers from two significant sports science HEIs in Romania to provide specialists with possible directions to improve educational management in the post-pandemic period.

### 1.1. Online Teaching in Higher Education

Online teaching has generated significant challenges for universities, which has resulted in studies analyzing its impact on the efficiency of the teaching process. In addition to its benefits for universities, which had the opportunity to attract large numbers of students from different geographical areas, the new concept maps of online teaching imposed by the COVID-19 pandemic have changed the perspectives, behaviors, and social engagement patterns of managers, teachers, and students. Some challenges were addressed to the university management and IT specialists who, in some institutions, did not properly support the teaching staff [1]. Others aimed at preparing teachers for online teaching, which, given the general use of IT in education, managed to ensure the prerequisites for the continuity of the teaching process [5].

Hodges et al. (2020) speak about Emergency Remote Teaching (ERT), which they define as the "use of fully remote teaching solutions for instruction or education that would otherwise be delivered face-to-face or as blended or hybrid courses" [8]. Such an approach highlights the response of the education system to the crisis caused by the pandemic and reveals the specific difference from online teaching, namely that teachers were not prepared for this system, but they adapted as a result of the restrictions imposed on face-to-face teaching activities. ERT evaluation in pandemic conditions involves: evaluating the context of the teaching process (internal and external resources related to the technological infrastructure), input evaluation (participant responsibility expressed through feedback from teachers, students in connection with the implemented action plan), evaluating the teaching process (monitoring participants, evaluating training methods, problems faced by participants, flexibility of the teaching process), service evaluation (feedback from participants on learning outcomes). The authors emphasize that the ERT model evaluation should focus more on the first three components than on learning outcomes.

Using factor analysis [9], another model was created on the efficiency of the teaching process, including (F1) school management support, (F2) family–work conflict, (F3) home

infrastructure, and (F4) technology choice. Of these four factors, the first two have the highest level of correlation with the teaching process [9]. However, the other two factors can have a great impact on the efficiency of the learning process.

For a better understanding of the topic, we organized the information taking into account these two-factor models and detailed the content by analyzing the literature relevant to our research.

The distance learning system based on video conferencing is recognized as a modern online teaching method in the 21st century, an environment that facilitates bidirectional interaction, collaborative learning, but also a similar level of satisfaction as the face-to-face approach [10]. In this way, an essential factor in stimulating students' interest in online learning is the university's potential to provide synchronous virtual classroom environments accessible with smartphones and laptops from home in optimum connection conditions [11].

Universities carry out online learning or short e-learning activities using various teaching management systems (Adobe Connect, Academic Learning Management System, Google Meet, Microsoft Teams, Perculus, Zoom, etc.) according to their capacity and technological infrastructure for the distance education process [12].

Distance learning and MOOCs (massive open online courses) are very useful tools for sustainable higher education. Provided through e-learning, online teaching methods are alternative solutions to continue the activity in HEIs during the pandemic. Nowadays, e-learning is not only a trend but a necessity. Online videos, synchronous meetings, asynchronous messaging and course studying, augmented reality (AR) books and virtual reality (VR) simulations are preliminary steps in the process of learning physical education and sports via the online environment. VR simulations have proven to be effective solutions in times of crisis (such as during the COVID-19 pandemic) or as additional training for sports activities [13,14]. They can be a valid solution in the post-pandemic period.

Online education also poses a number of obstacles to teachers' ability to use different teaching methods, involves less coverage of the curriculum content, and some students lack the required technological skills, which threatens the learning progress [15].

The most common barriers to online educational activity were the lack of internet access and the fact that students were not accustomed to this type of activity [16]. Social media applications such as WhatsApp, Google Classroom, Zoom, and e-Learning were among the most used platforms by universities. The various materials were mainly delivered through WhatsApp messages or other social networks. A problem has been identified in this regard, namely the ability of mobile phones to store all information.

In these circumstances, the focus was on several goals of HEIs, such as: (1) addressing teacher and student skill gaps and mismatches, and (2) increasing the attractiveness of the curriculum and reforming it by including real-world applications, inquiry-based and ICT-enriched learning, and collaborative practices [17].

Technological adaptation is considered to be the most important advantage for the young generation during the transition period [18]. Some authors state that most students attend online/distance education courses with their smartphones, while others claim that using devices to access the distance education system affects the level of student satisfaction, and the average satisfaction score of students who can access the internet with a device increases significantly [11,19]. Thus, according to the distance education survey conducted in Turkey by the Council of Higher Education (CoHE) [20] regarding the pandemic process, 83% of students reported having electronic devices that allowed them access to distance education, and 97% of them stated they had sufficient internet access. Other studies emphasize that the technological equipment of students is the basis of efficient teaching, together with their ability to use technology, their perceptions, and attitudes towards online teaching [21].

Another paradigm that makes an important contribution to understanding the topic is actors' participation in the learning process: teacher, student, and teacher/student relationship. Teachers are vectors able to support the effort of promoting the inclusion of

IoT (Internet of Things) in the education system, given their role in the learning process. Thus, teachers, who are primarily responsible for integrating the e-learning system into higher education, have indicated that distance/digital learning streamlines teaching, and develops their practical computer skills. In their opinion, digitized courses on various platforms can both ensure professional development and rejuvenate universities. Teachers have to face the action of external factors such as equipment, resources, level of training, and technical support, but also internal factors such as the level of IT knowledge and skills, and attitudes towards the implementation of technology in the teaching process [22]. At the same time, they should respect the opinions of students, stimulate their creativity, and develop their entrepreneurial spirit. Teachers also need to be available to empathetically interact with their students outside of virtual classrooms. Moreover, when teachers carefully structure the pedagogical design units, students will become independent, critical, and organized [23,24]. They will navigate confidently in the "sea of knowledge" without being disturbed by the influx of information.

Teachers will reconsider the students' open-mindedness to interchangeable roles, teaching role—student roles. Students are sensitized to create a teacher-student "balance of power" and, by harmonizing these roles, they will be able to support the idea of "democratization of education" [25]. In the context of distance learning, students will be able to manipulate objects in the virtual environment, communicate coherently and in real-time in this environment, as in the real world, and live life like-experiences created by simulated scenarios [26]. Teachers should pay more attention to their interactions with students, despite the distance imposed by the online learning process [27–29].

Some authors emphasize the role of cooperative skills in online learning based on teacher instruction and group cooperation by discussing and encouraging group members to participate actively. They also state that cooperative skills can be developed by setting tasks for each member of the group and fostering communication between them [30]. Academic challenges are great, but they can change negatively when the individual (visual, auditory, tactile-kinesthetic) learning style is not respected and the teaching materials are not "communicative", namely, they are not accessible to all, or when these learning aids do not trigger attractive, multi-, inter-, or transdisciplinary academic collaborations.

Even through online education individuals develop their cognitive ability (by assimilating knowledge, skills, time management notions, metacognition, etc.) and affective behavior (which includes motivation, emotions, procrastination, self-esteem, etc.). Moreover, individual motivation and confidence are crucial for learning competencies [31]. During virtual studies, new digital strategies can ensure the meaning and success of the instructive-educational process, but also the active participation of students. In this modern and flexible educational context, students are also motivated by:

— the importance of advanced mass data processing technologies in increasing learning efficiency;
— the possibility of acquiring general skills necessary for any profession—active learning, active listening, and teamwork;
— open, comfortable dialogue that "breaks the ice" of reality;
— multi-, inter-, or transdisciplinary academic collaborations;
— diversified learning experiences;
— enriched emerging ideas, etc.

But to assure a high-quality teaching process, teachers should have their own state of comfort and well-being, which refers to the self-efficacy belief that teachers should have in their activity. Some studies emphasize that educators should reorganize their teaching style by working a lot in their free time to prepare online lessons [24]. Moreover, even though they seemed to accept these forms of the teaching process, more than 50% reported physical and mental discomfort as a consequence of excessive online activities [23].

Most conclusions show the contribution of the COVID-19 pandemic to the "reinvention of university teaching" [25] by transforming the teaching process in accordance with online system requirements [22].

*1.2. Sports Science HEIs in Romania—A Pre- and Post-Pandemic Approach*

Sports Science is a field with a strong interdisciplinary component, which examines how the human body works, develops, and interacts to achieve sports performance.

In Romania, university curricula in the field of sports and physical education are aimed at providing both the initial training of specialists through bachelor's programs and specialization through master's, doctor's, postgraduate, and postdoctoral programs. The way of delivering these programs is mostly traditional. In sports science programs, e-learning platforms are used for part-time education, as support for full-time education, or as a training environment for continuing education programs [14].

The standards underlying the preparation of curricula for bachelor's programs provide (out of the total number of 4500 h corresponding to the 180 credits) for a maximum number of 50% credit units earned through individual study activity. The standards also provide for a 1:1 ratio between the number of course hours and the number of practice hours (seminars, laboratories, projects, specialized practical training, etc.), with a permissible deviation of $\pm20\%$, which indicates that practical activities are an important component [32].

The pandemic has generated several problems related to the way of approaching traditional face-to-face activities (provided as a basis for student activity) in an Emergency Remote Teaching system provided in an online or hybrid (mixed) format. In this context, the way of performing individual activities has also changed, the reduced possibility of student interaction with different sports groups leading to a reconsideration of the methods and means used in the teaching-learning process [33,34].

Essentially, online activity in the field of sports science involved a reduction in physical interaction between teachers and students but also between students and community members, which has led to changes in the practices of admission to study programs in the field, the methods of knowledge transfer to students and the way of examining students through mostly theoretical and less practical knowledge (at least in the first year of the pandemic), but also to reduced access to learning resources and therefore limited research activity caused by the impossibility to interact with target groups [35,36].

Emergency Remote Teaching activity in the field of sports and physical education science was based on the previous experience of European and non-European universities as regards the way of conducting study programs without physical presence. Online physical education poses a unique set of challenges in translating traditional physical education into a digital space, all while respecting the same benchmarks, curriculum, and evaluation standards as for traditional courses [37]. In the context of COVID-19, careful considerations should be made by physical education teachers in HEIs to ensure that their programs can maintain the learning outcomes proposed for pre-service teacher development at individual and social levels [38]. According to some authors, student experience and graduate learning outcomes may be hampered if research, practice, and policy fail to connect initial online teacher training [39].

Research conducted before the pandemic [37,40] identified important factors in carrying out programs in the field of sports and physical education science: modeling online educational practices, ensuring teacher-student interaction, transitioning online pedagogical and content knowledge, and navigating instructional tools and technology.

Following the impact of the pandemic, a need for supporting university teachers to also develop other skills, beyond the academic component, is identified [41], and the potential now exists for universities to innovate and improve their technological infrastructure by scaling up training for educators and upgrading emerging technologies [42].

A real threat in the field of sports science is represented by the uncertainty that exists around the new university standards, the production and delivery of engaging materials, the poor academic preparation, and the overall sustainability of physical education teacher education programs [38]. Reluctance to use online education ranged from student accountability, course rigor, safety and liability issues, retention rates and, most noted, the concern about how students would achieve national and state-level content standards and outcomes [43–45].

Specialists believe that reaching the new generation of learners in online and distance formats requires greater access for educators to tools and technologies able to facilitate content organization, delivery, and interactive modalities [37].

Addressing hybrid or online study programs, some authors have shown that students perceive the source of their satisfaction and academic achievement based on their ability to determine the pace at which they complete distance education courses and the strong online presence of the instructor [45]. Students in the above study reported the need for communication between the instructor and the student and the clarity of the instruction. Issues that need to be considered to ensure quality online education are interaction with students, meeting learning requirements and needs, necessary infrastructure, opportunities for the staff to use distance education tools, possible difficulties encountered by students and teachers, outcomes and performance achieved upon completion of the study program, feedback from students and teachers [46].

## 2. Sustainable Education in Sports Science HEIs through IoT

Starting from the challenges faced by sports science HEIs, one can state that sustainable education is facilitated by interconnected mobile technology (IoT—Internet of Things) and the adaptation or redefinition of existing curricula after reconsidering the competencies required by sports science programs. The pedagogical background is the proposed Internet-of-Things supported collaborative learning (IoTSCL) paradigm, which is based on constructivism and provides a highly motivating learning environment in university by promoting collaboration between students and creating new knowledge in a reflective process guided by the teacher [47]. IoT in higher education provides the learning process access to data coming directly from a device mounted to the student's asset from the institution or home based on intelligent, cross-platform software solutions.

Sustainable education can be achieved by adapting the content of taught disciplines and introducing new disciplines, which results in long-term sustainable development. In this regard, significant investments are needed to improve teachers' knowledge and personal development, which will allow them to adapt to new technologies and find new motivational strategies for students [48,49].

Therefore, the learning content that needs to be adapted involves the following aspects [50]:

- Where they are: if they are at the office, on campus, traveling, etc.; if they have spare time, students should be able to access the learning content remotely as a podcast, YouTube video, or extended reality (XR) application;
- When they can access the information (a very flexible schedule, 24 h a day);
- How they can assimilate it (written text is highly valuable, but it can be more easily understood if associated with photos, videos, simulations, applications, etc.).

More than ever, in times of crisis (such as during the COVID-19 pandemic), the IoT-enriched learning environment was the solution for continuing the process of training specialists in the field of sports science. For physical education and sports students, the intention to use synchronous virtual classroom environments is positive, and most of them have sufficient cognitive devices and internet infrastructure to participate in synchronous virtual classroom lessons [19].

### 2.1. IoT-Enriched Learning Environment in Sports Science

Taken from the field of e-learning, many teaching methods were the alternative for continuing the activity in HEIs during the pandemic. A practical field par excellence, physical education, and sports are favored by the development of sports training methodology concerning ICT, in general, and IoT, in particular [51]. Currently, technological advancements have expanded the boundaries of training opportunities in an interdisciplinary field.

Today, athletes can use VR glasses for an intense sense of reality when performing (in simulated conditions) different motor actions specific to sports such as [52–54]:

— Tennis, baseball, badminton, volley, in which case training takes place in special rooms where the VR system simulates ball throwing in accordance with a software parameter—intensity (beginner to advanced), strength, speed, trajectory, tricks, etc.;
— Team sports (basketball, football, rugby, etc.), which are played in a VR environment where athletes can take different roles;
— Skiing, water-skiing, skateboarding, fitness, swimming, which are performed using simulators or an XR environment, the application provides constant feedback for errors.
— Tai-Chi, Qigong, Yoga are very appropriate for VR because they involve 360-degree movements, and students cannot see the Sensei's movements all the time in the real world. The examples can continue.

Over time, IoT has provided very useful instruments for teaching physical education and sports, enhancing the learning curve and teaching efficiency.

In sports science, the learning process can be much developed through Hi-tech multimedia support, which involves attractive or dynamic charts, graphs, maps, video lectures, and data visualization. Thus, interactive elements will revive the field of sports science.

### 2.2. New Premises in Applying IoT to Sports Science for the New Generation of Students

Decentralization is achieved through connectivity, and young people are extremely attracted to the concept of IoT interaction, especially in the learning process. Connectivity learning is based on four primary principles: "autonomy, diversity, openness, and connectedness/interactivity" [51,55–57].

Students are very good at using mobile devices and technology and see life through IT lenses. They cannot be taught like other generations through books and printed courses, because they want to try things, simulate, hear, feel, sense, taste, etc. [58].

They are more innovative and creative than the older generation because they try to immerse into the phenomenon and understand it better. VR, the stimulus connected to the body, and the application's constant feedback lead students to learn the characteristics of motor skills and test different postures/movements without damaging their health [58–60]. Students' imagination is stimulated by the activities performed in the XR environment.

VR and AR in education are associated by students with deep comprehensive learning due to the movement reproduction ("learning-by-doing effect") and the involvement of all senses during simulated reality [56,61]. Difficult topics can be taught and facilitated by VR technology through interactive and interesting presentations and constant feedback [57,61].

IoT is no longer a trend but an everyday tool that enables teachers and trainers to uninterruptedly monitor students/athletes who perform sports activities due to its IT components (accelerometer sensor used to collect motion data, magnetometer sensor, skin conductance sensor, wrist pulse oximeter sensor, and temperature sensors are used to monitor oxygen level, pulse rate, and temperature during sports performance, etc.), which are attached to the athletes' bodies. The collected information is transmitted by a wireless internet network to a server where appropriate applications are installed to analyze and interpret the data. The data are also collected through visual sensors such as Kinect sensors, infrared cameras, etc. [62,63]. Thus, the teacher and the trainer can constantly provide appropriate feedback to students to help them improve their execution and performance. Visual sensors are not very suitable for outdoor activities because they cannot provide enough details. Kinetic sensors offer accurate information on students' performance (such as movement/contraction intensity, timing, bio-physiological coordinates, range of motion, etc.) [64]. Such detailed and important information helps teachers and trainers to adapt their strategies and set new goals, teaching/training methods, hygiene programs, and physical support [62–64]. Constant monitoring and the appropriate adaptation of the sports science program help athletes to highly develop "accuracy ratio (98.3), prediction ratio (96.5%), interaction ratio (94.4%), performance ratio (95.1%), the efficiency ratio (93.2) and reduce error rate (17.5%), and physical activity patterns" [51].

Another characteristic of IoT is that the applications that analyze and interpret data are based on artificial intelligence (AI) and machine learning algorithms stored in the cloud through the IoT module. The server that makes the inferences will send the results to the teacher's and trainer's mobile phones. The application learns the student's behavior, characteristics, and skills and finds out new methods to challenge their limits. Through this interaction, both students and the application progress together, helping each other, overcoming challenging boundaries. It is interesting that new students can benefit from the application's accumulation under the constant monitoring of teachers and trainers and can help new students to develop their physical skills. At the same time, the application responds with an individualized program for each athlete due to the machine learning algorithm after getting to know the athlete [62,65,66].

Studies show that correctly practiced physical activity has a strong positive influence on students' health, boosting their academic performance. Learning becomes easier and more fun, and students' motivation, understanding, and deep learning increase very much [62]. Through sports-dedicated applications stored in the cloud and accessed through IoT, the evaluation of students and athletes is highly accurate and provides teachers and trainers with the appropriate tools to help improve athletic performance.

Using IoT, devices interconnected into a cloud platform and mobile devices as user interface, VR/AR applications dedicated to physical education and sports can be real support for teachers, trainers, students, and athletes. The data stored in IoT is brought to students/athletes by modeling the 3D scene, VR scenes, real-time video stream, etc., rendered on mobile devices. "The Internet of Things platform is responsible for data collection and real-time control of the virtual scene" [65].

In sports science, it is generally difficult to isolate the key factors of performance (be they biomechanical, physiological, or psychological) and analyze them separately. For this reason, specialists propose VR that can overcome these limitations and can compensate for the lack of sports arenas [66].

As regards the training of specialists employing IoT in a changing dynamic field such as sports science, it can be stated that young people's adaptation to working with various digital gadgets is easily achieved. The young generation uses mobile devices all day long and everywhere. Young people use smartphones, consoles, tablets, laptops, and smartwatches to transmit information quickly and easily. "They are eager to read brief and limited content or watch short videos instead of listening to long lectures" [67].

Students prefer smartphones and tablets in a digital classroom, which develop their technical expertise. The explanations of complex sports strategies and biomechanical or physiological exercise parameters via IoT can provide a better understanding. Through this wireless network of various devices, learners can keep an online real-time track record of their athletic progress on their portals or can use interactive graphics. They can also exploit a haptic space to develop detailed perceptions of the movement. Contextually, the pseudo-haptic objects that they can virtually guide and whose movement is imposed by the computer, depending on exercise dosage or a particular technical task, will support sports students to perform technical training or exercise individually during practical sessions.

IoT can improve the learning process, and especially VR solves the problem of the separation between perception and action, making them more natural [65–68]. Moreover, IoT raises significant challenges that could solve problems related to:

— The monotony of a traditional course—the learning process can progress by combining live classes with pre-recorded lectures and online courses;
— Changing the pedagogy paradigm when the focus on knowledge and learning by memorizing information converges towards the student-centered approach;
— Training individualization—teaching will take into account student's acquisitions;
— Differentiation of teaching units—by creating an autonomous and flexible learning framework for each student to follow their path, although it belongs to a collective approach (Key data on education in Europe 2012, online);

— Transforming the didactic triangle (teaching, learning, training) from the perspective of the actions specific to a learning community including teachers and learners [63,64,69];
— Securing interaction in a "hyperconnected world", etc.

## 3. Methodology and Results

### 3.1. Methodology

A complex survey was designed to find out about the experiences of teachers during the COVID-19 pandemic in Romania. A Google Forms Survey was administered to sports teachers by the deans of the National University of Physical Education and Sport in Bucharest (UNEFS) and Spiru Haret University in Bucharest (USH). The data were analyzed using quantitative methods to understand the challenge faced by physical education and sports teachers in HEIs. The survey contains open- and closed-ended questions to gather information from teachers on the alternative methods used and the adaptation of their teaching during this period. Multiple-choice questions were also provided.

#### 3.1.1. Sample

Research participants were physical education and sports teachers from higher education institutions in Romania. The study was conducted between 5 March and 20 June 2021. The statistical population is made up of 50 UNEFS teachers and 85 USH teachers, meaning 135 respondents in total. A representative sample with a 95% probability and 0.5 errors is 100 teachers who use different online teaching methods. The survey collected 100 responses, so it is representative and consists of three main sections:

— The first section emphasizes the psychological impact of working from home;
— The second highlights the online platform, applications, and methodologies used in teaching and learning;
— The third focuses on future adaptations needed in sports science HEIs.

#### 3.1.2. Methods

Most questions were rated on a 5-step Likert scale, where step 1 means the lowest impact, 3 means a neutral impact, and 5 means the highest impact. These data were the basis of our inferential statistics using SmartPLS.

#### 3.1.3. Hypotheses

The research hypotheses are:

**Hypothesis 1 (H1).** *ERT applied to the field of Sports Science significantly impacts the teacher's psychological state.*

**Hypothesis 2 (H2).** *ERT brings significant changes in the lifestyle of teachers in HEIs.*

**Hypothesis 3 (H3).** *Sustainable university decisively determines the methods that positively influence e-learning.*

**Hypothesis 4 (H4).** *Methods such as challenge-based learning, solving real problems within small groups, inquiry teaching, and experiential learning through simulations and XR will positively influence students' transversal skills (collaboration, communication, creativity, and critical thinking).*

### 3.2. Research Results

STEP I—In the first step of our analysis, we run an inferential analysis to test the first two hypotheses.

To verify these hypotheses, we analyze 3 reflective variables: Limits (limitations brought by technology) (Table 1), Mental (psychological impact of online activity on teachers) (Table 2), and Balance (work-life balance during the pandemic) (Table 3 and Figure 1).

**Table 1.** Limits (negative effects) associated with working from home.

| Components of the Limits Variable | % |
|---|---|
| Interaction with students was superficial | 62.7 |
| Student activity or involvement was reduced | 61.4 |
| House could not offer the teaching atmosphere from university | 60.2 |
| Social contacts were limited to family members | 53 |
| Technical problems during teaching | 44.6 |
| Increased internet and energy costs | 27 |
| Professional marginalization | 28.9 |

**Table 2.** Psychological impact on teachers.

| Components of the Mental Variable | % |
|---|---|
| Positive influences | |
| Flexibility | 54.2 |
| Safety/security | 53 |
| Pleasure | 20.5 |
| Freedom | 20.5 |
| Negative feelings | |
| Loneliness | 34.9 |
| Boredom | 25.3 |
| Frustration | 22.9 |
| Lethargy | 14.5 |
| Outrage | 15.7 |
| Depression | 9.6 |
| Anger | 4.8 |

**Table 3.** Latent variable correlation (Limits and Mental).

| | Balance | Limits | Mental | Results |
|---|---|---|---|---|
| Balance | 1.000 | −0.357 | −0.466 | −0.124 |
| Limits | −0.357 | 1.000 | 0.776 | 0.551 |
| Mental | −0.456 | 0.776 | 1.000 | 0.387 |
| Results | −0.124 | 0.551 | 0.387 | 1.000 |

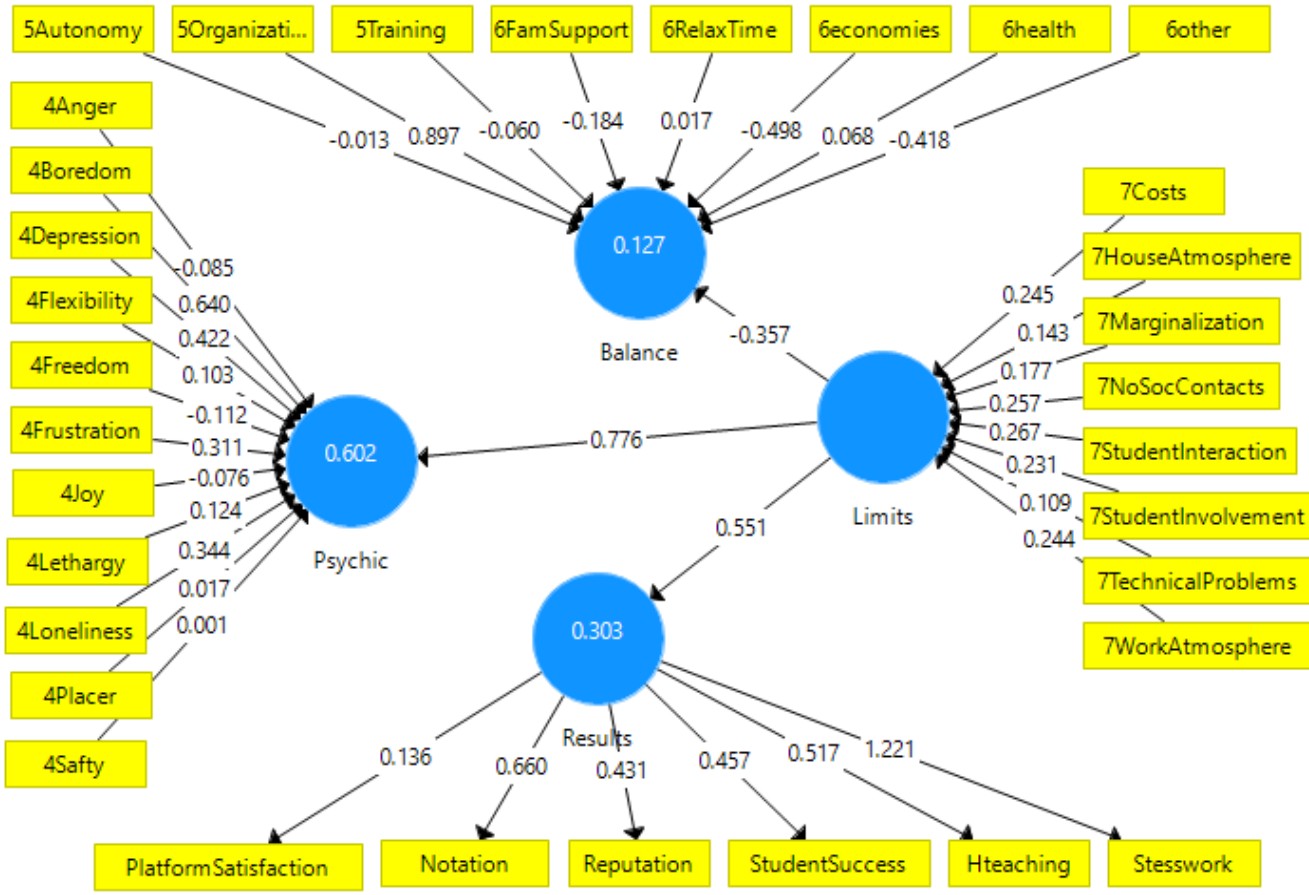

**Figure 1.** The influence of online teaching limitations on teachers' work-life balance, psychological balance, and obtained results.

### 3.2.1. The Psychological Impact of the ERT System on Teachers

Our analysis reveals a strong positive correlation between Limits and Mental variables (0.776), meaning that limiting online teaching has a strong effect on teachers' psychological determination (Table 3).

### 3.2.2. ERT Impact on the Lifestyle of Teachers in HEIs

Working online with students during the COVID-19 pandemic was more stressful than traditional teaching for most educators (34.9%), with 19.3% saying that it was very stressful. They seem to have found appropriate teaching solutions, as 42.2% were pleased and 26.5% were very pleased with the results of their online work during the COVID-19 pandemic. Teachers were able to maintain a good work-life balance (26.5%—good balance and 34.9%—very good balance). Other positive effects associated with working from home are shown in Table 4 and Figure 1.

**Table 4.** Positive effects associated with working from home.

| Variable | % |
|---|---|
| Possibility of additional online training | 59 |
| Better organization of professional activity | 49.4 |
| Increased decision-making autonomy | 38.6 |
| Saving time and money (did not go to work) | 73.5 |
| Taking care of one's family and health | 43.4 |
| Supporting other family members | 33.7 |
| Having more time for relaxation | 22.9 |

The fourth variable is formative and refers to the overall impact of online activity during this period, including teacher satisfaction with the platform (Loading factor = LF = 1.336), correct grades during the examination process (LF = 0.66), student success (LF = 0.457), number of teaching hours (LFA = 0.517), level of stress during online work (LF = 1.221), and university reputation (LF = 0.431). A very important remark is that the overall results of online teaching were good (as evaluated by LF), but with the cost of a very stressful online activity (1.221) (Figure 1).

A medium positive correlation was observed between Limits and Results (0.551), meaning that the overall results of online teaching were good, but affected by these limitations and with a very high cost of stress. A medium negative correlation was noticed between Limits and Balance (−0.357), meaning that a new work-life balance was established, but with implications for teachers' personal lives. A medium positive correlation was observed between Mental and Results (0.387), meaning that the overall psychological impact influences the online teaching results. As regards Mental and Balance (−0.466), the score indicates that the overall psychological impact affects the teacher's work-life balance. This interpretation is supported by the values of Path Coefficients (Table 3).

We can observe in Tables 5 and 6 that all the indicator criteria (Composite Reliability, Rho_A, Cronbach's Alpha, and AVE) are calculated only for the fourth variable because it is a formative one. They are all above the critical limit, meaning that they meet the criteria, and our model is consistent, as the reflective variable indicators are not calculated.

**Table 5.** Path Coefficients.

| Matrix | Balance | Mental | Results |
|--------|---------|--------|---------|
| Limits | −0.357 | 0.776 | 0.551 |

**Table 6.** Model validation criteria.

| Variables | Composite Reliability | Rho_A | R-Square | Cronbach's Alpha (CA) | AVE |
|-----------|----------------------|-------|----------|----------------------|-----|
| Minimum limit | >0.7 | >0.7 | >0.5 | >0.7 | >0.5 |
| Balance | | 1 | 0.127 | | |
| Limits | | 1 | | | |
| Mental | | 1 | 0.602 | | |
| Results | 0.775 | 0.775 | 0.303 | 0.853 | 0.49 |

Discriminant validity: Our model is statistically powerful, as the Fornell-Larcker criterion is met (Table 7): Limits -> Mental (0.776), Limits -> Results (0.551), Mental -> Results (0.387).

**Table 7.** Fornell-Larcker Criterion.

| | Balance | Limits | Mental | Results |
|---|---------|--------|--------|---------|
| Balance | | | | |
| Limits | −0.357 | | | |
| Mental | −0.466 | 0.776 | | |
| Results | −0.124 | 0.551 | 0.387 | 0.659 |

The SRMR and d_ULS values for the estimated model (0.121 and 8.218) are greater than the SRMR and d_ULS values for the saturated model (0.119 and 7.934). Thus, we can state that our model fits H1 and H2, so the hypotheses are accepted (Table 8).

**Table 8.** Model Fit.

|  | Saturated Model | Estimated Model |
|---|---|---|
| SRMR | 0.119 | 0.121 |
| d_ULS | 7.934 | 8.218 |

The variance inflation factor (VIF) of each construct was calculated to check the significance of variables. Table 5 shows an overview of the findings. VIF is much lower than the maximum limit accepted (5), meaning that no collinearity is manifested between variables. Based on the above criteria, we can state that H1 and H2 are accepted (Table 9).

**Table 9.** Overview of the VIF findings.

| Variable | VIF | Variable | VIF | Variable | VIF |
|---|---|---|---|---|---|
| 4Anger | 1.177 | 5Autonomy | 1.113 | 7Costs | 1.239 |
| 4Boredom | 1.495 | 5Organization | 1.147 | 7HouseAtmosphere | 1.617 |
| 4Depression | 2.815 | 5Training | 1.201 | 7Marginalization | 1.140 |
| 4Flexibility | 1.272 | 5FamSupport | 1.213 | 7NoSocContacts | 1.368 |
| 4Freedom | 1.825 | 6RelaxTime | 1.088 | 7StudentInteraction | 1.471 |
| 4Frustration | 1.667 | 6Economies | 1.211 | 7TechnicalProblems | 1.310 |
| 4Joy | 2.085 | 6Health | 1.256 | 7WorkAtmosphere | 1.486 |
| 4Lethargy | 2.354 | 6FamSupport | 1.213 | Hours of teaching | 1.760 |
| 4Loneliness | 1.204 | 6RelaxTime | 1.088 | Rating | 1.745 |
| 4Safty | 1.530 | 6Economies | 1.211 | Platform Satisfaction | 2.260 |
|  |  | 6Health | 1.256 | Reputation | 3.851 |
|  |  | 6Other | 1.142 | Student Success | 4.824 |
|  |  |  |  | Stress Work | 1.505 |

STEP-II—in the second step of our research, we developed an inferential analysis to check the Hypotheses 3 and 4 of our research. The analyzed variables are shown in Table 10.

**Table 10.** ID and definition of variables analyzed.

| Variable | ID | Definition |
|---|---|---|
| 2ResultContent | 2 | I am satisfied with the results of my work in the online teaching process during the COVID-19 pandemic. |
| 10PlatformSatisfaction | 10 | Express your satisfaction with the platform(s) you used for video and audio communication with students and colleagues |
| 11DigitalSkills | 11 | I have significantly improved my digital and other technical teaching skills required for post-COVID period. |
| 12Commitment | 12 | I think that working from home could be my permanent commitment to teaching. |
| 14CriticalThinking | | |
| 14Collaboration | | |
| 14Communication | 14 | Student Competencies: I think that the important items of online teaching (e-learning) are: |
| 14Creativity | | |
| 14TechnicalSkill | | |
| 14Environment | | |

**Table 10.** *Cont.*

| Variable | ID | Definition |
|---|---|---|
| 16ContentNOcompetencies | | Student Competencies |
| 16disciplineNOtransferableSkills | 16 | What is more important for students |
| 16STEAMNoProblemSolving | | to achieve? |
| 15Experiential | | Teaching methods: How important it is to use in e-learning: (a) experiential teaching and learning |
| 15Investigation | 15 | (b) interactive teaching and learning |
| 15Challenge | | (c) challenge-based learning |
| 15Questions | | (d) inquiry teaching |
| 17 MOOCsEL | 17 | Are MOOCs effective tools for e-learning? |
| 18VRARExperiential | 18 | Is VR teaching a good solution for experiential and gaming teaching? (https://teach-vr.com/platform/ accessed on 23 January 2022) |
| 22 How important do you think the following types of activities are for adapting educational institutions to the accelerated pace of skills in the labor market? | | |
| 22Exams | | Do you think that exams in the online session were comparable to face-to-face ones? |
| 22SustainableUniv | | Sustainable approaches aimed at new competencies in organizational management |
| 22Reputation | | Organization's reputation in terms of adapting the curricular area |
| 22Foresight | 22 | Academic foresight in the field of new trades |
| 22Mobility | | International academic mobility related to new skills and abilities adapted to the needs of the global labor market |
| 22Digitization | | Resource management for change adaptation objectives |
| 22StudentSuccess | | Student success in new skills |
| 22Curriculum | | Adapting the curriculum to new occupations and transversal competencies and challenges in the labor market |

The correlation coefficient shows a strong correlation between:

- Variable 2 with 10, 11, 12, 14, 15, meaning that teachers were more satisfied with the results of their work in the online teaching process during the COVID-19 pandemic. They were also satisfied with the online platform used. They have improved their digital and other technical skills. They consider a possible future commitment to teaching online. They believe that the important items of online teaching (e-learning) are collaboration, communication, technical skills, and the appropriate environment for teaching. These teachers rely on: (a) experiential teaching and learning (making experiments in natural environments or at least the virtual environment) (b) investigation—interactive teaching and learning (making up small groups to solve mini-tasks) (c) challenge-based learning (students need to find solutions for a current

problem/challenge such as COVID-19) and (d) inquiry teaching (asking questions as students to find out the content of the course by themselves)

- Variable 10 with 11, 12, 14, 15, 17, 22. Teacher satisfaction with the platform was in accordance with the elements presented above. Nevertheless, teachers considered MOOC platforms and VR applications for experiential and gaming teaching to be good solutions in the pandemic context. Teacher satisfaction was strongly correlated with major changes in the university development, such as: (a) sustainable approaches aimed at new competencies in organizational management (creating and developing an entrepreneurial attitude among students and improving their ability to cooperate with entrepreneurs and companies from the country and from abroad, partnerships with the business environment, clusters with the active involvement of teachers and students); (b) organization's reputation in terms of adapting the curricular area (satisfaction level of the beneficiaries, level of national and international evaluation of the organization); (c) international academic mobility related to new skills and abilities adapted to the needs of the global labor market (number and outcomes of academic mobility programs) and participation of the organization in RDI programs with applications to the development of competencies); (d) resource management for change adaptation objectives (investments in online platforms, hybrid educational platforms; investments in digital transformation (technology transfer centers, clusters, regional university consortia, digital hubs); (e) student success in new skills (number of students participating in research activities in disruptive fields and technologies, who are enrolled as business accelerators, start-ups; insertion of graduates in the labor market), and (f) adapting the curriculum to new occupations and transversal competencies and challenges in the labor market (inclusion of the digitization component throughout the educational process; distinct digitization chapters for each study program; number of inter- and transdisciplinary study programs.
- Variable 11 with 12, 14, 15: teachers have improved their digital skills during the pandemic, which positively affects the teaching process with all its features: commitment, experiential, investigation, challenge, questions, critical thinking, collaboration, communication, creativity, adequate environment.
- Variable 12 with 14, 15, 17, 22: it seems that these major curriculum changes, the advantages brought by MOOCs, and the new teaching methods positively correlate with the future commitment of teachers to the online environment.
- All items of variables 14 and 15 strongly correlate with all items of variables 17, 18, 22. We can assume that these new efficient teaching methods are directly related to the major transformation in universities due to the adoption of technology.
- There is a very strong correlation between the adoption of MOOCs and VR/AR experiential teaching and the opportunities brought by the digitization of technology.

### 3.2.3. Construct Reliability and Validity

SmartPls software provides many tests that can be used to ensure a coherent analysis and interpretation of data and to assume the research outputs. For example, the consistency of our model was grounded on the validation steps provided in Table 1 [70,71]. All the considered variables have very high values for composite reliability. Cronbach's Alpha and Rho_A (>0.7—the bottom value authorized) and average variance extracted (>0.5—the bottom value authorized (Table 11).

**Table 11.** Validation criteria of the structural equation model.

| Variables | Sub-Items | Loading Factor (LF) | AVE | Composite Reliability | R-Square | Rho_A | Cronbach's Alpha (CA) | VIF |
|---|---|---|---|---|---|---|---|---|
| | Minimal Limit | >0.6 | >0.5 | >0.7 | >0.5 | >0.7 | >0.7 | |
| Competencies | 14Collaboration<br>14Creativity<br>14CriticalThinking<br>14Environment<br>14TechnicalSkill | 0.863<br>0.947<br>0.820<br>0.835<br>0.863 | 0.751 | 0.938 | 0.973 | 0.940 | 0.937 | 3.847<br>5.011<br>3.343<br>3.012<br>2.863 |
| Methods | 15Experiential<br>15Investigation<br>15Questions | 0.839<br>0.914<br>0.889 | 0.777 | 0.912 | 0.684 | 0.914 | 0.911 | 2.506<br>4.040<br>3.623 |
| Teaching | DigitalSkills<br>VRARExperiencial<br>Simulation<br>FairRating<br>ResultContent | 0.740<br>0.811<br>0.305<br>0.554<br>0.732 | 0.428 | 0.775 | 0.977 | 0.818 | 0.758 | 2.237<br>1.433<br>2.177<br>1.257<br>1.921 |
| Univ | 22SustainbleUniv<br>22Exams<br>22Mobility<br>22Curriculum | 0.926<br>0.439<br>0.945<br>0.946 | 0.710 | 0.901 | | 0.945 | 0.886 | 5.017<br>4.223<br>1.323<br>5.069 |

Notes: The LF Univ does not have an $R^2$ value as it precedes the other variables in the SEM (structural equation modeling).

To verify the validity of the convergence, the average variance extracted (AVE) of each latent variable was calculated. All AVE values of the indicators analyze important factors (Univ = 0.71, Methods = 0.77, Competencies = 0.75) for the adaptation of universities to the accelerated pace of competencies required in the labor market. Thus, the following appropriate teaching methods and important factors in distance learning are higher than the acceptable threshold of 0.5, so the convergent validity is confirmed. There is an exception regarding the development of future teaching where the value is very close to 0.5 (Teaching = 0.428), but we do not know yet all the future trends in Physical Education and Sports (PES) training, such as artificial intelligence—AI, machine learning—ML, blockchain, big data, etc. (Table 11).

To extract the future directions for the curriculum development in sports science universities, teachers were invited to identify the main trends able to ensure the adaptation of higher education to the future labor market. The results are shown in Table 11 and Figure 2:

1. Sustainable approaches aimed at the new competencies in the organizational environment (creating and developing an entrepreneurial attitude among students and improving their ability to cooperate with entrepreneurs and companies from the country and from abroad, business partnerships, clusters with the active involvement of teachers and students) (LF 0.929).

2. Examination of future approaches with an emphasis on gaining competencies rather than content, on transferable skills rather than the number of subjects, studied, on problem-solving rather than learning by heart (LF 0.439). Anyway, this is an issue in constant development.

3. International academic mobility related to new competencies and abilities adapted to the needs of the global labor market (number of academic mobility programs, results of academic mobility), participation of the organization in RIA with a focus on developing field-specific competencies—LF 0.945), adaptation of the curricular area to new cross-cutting professions in the field of physical activities, new competencies and challenges in the labor market (inclusion of the digitization component throughout the educational process; distinct digitization chapters for each study program; the number of inter- and transdisciplinary study programs).

4. Adaptation of the curriculum to the new market and evolution of society is another item that has a high impact on the university's high standards (LF 0.945).

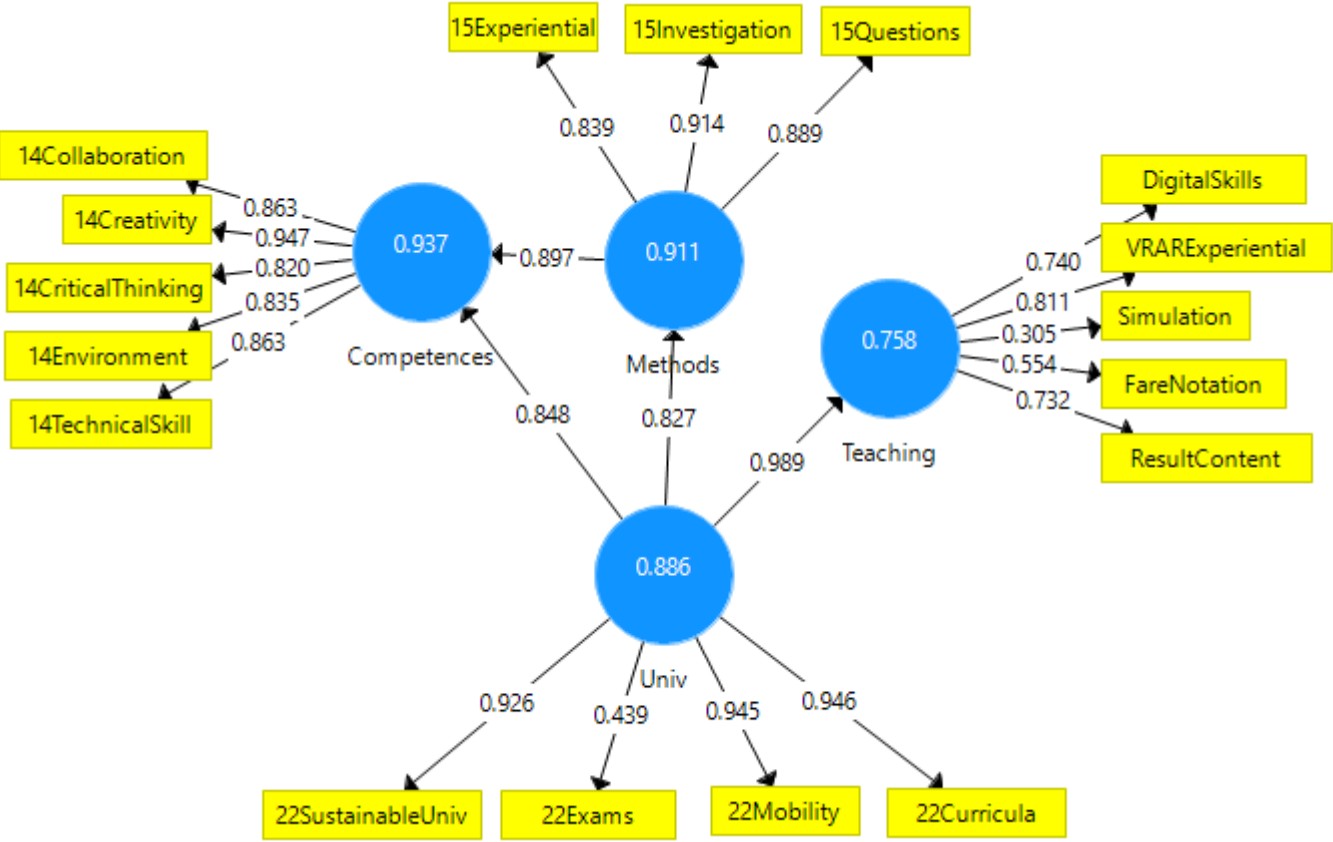

**Figure 2.** Cronbach's Alpha for 4 variables: Univ, Competencies, Methods, Teaching.

These 4 elements form the Univ variable, which also has a very high CA (0.710). The University's High-Quality Standards (Univ variable) are also influenced by university resources, student success with new competencies, academic foresight and the university's reputation, but we do not include these sub-factors in the model due to the multicollinearity effect.

University resources for the adaptation of objectives refer to investments in hybrid online educational platforms; investments in digital transformation - technology transfer centers, clusters, regional university consortium, and digital hubs. Student success with new competencies refers to the number of students participating in research activities, who are enrolled as business accelerators, start-ups; graduates' entry into the labor market. Academic foresight in the field of new trades concerning physical activities, health, and wellbeing areas are other factors that should be taken into account in the near future. The university's reputation in terms of adaptation of the curricular area refers to the level of satisfaction of the beneficiaries and the level of national and international evaluation standards.

We can observe that teachers would like to use in future Simulation Software (LF 0.305), Virtual Reality and Augmented Reality for experimental teaching (LF 0.811), and e-learning platforms only as assistive technologies and fair rating (AVE 0.554). In the teaching process, the digital skills of teachers are very important (LF 0.740), as they claim that overall, they are content with the results obtained during the pandemic in the online environment as a crisis solution (LF 0.732), but they prefer teaching in the physical environment. These 5 sub-items form the Teaching variable of our model, which refers to teachers' open-mindedness toward new technologies and digital skills (CA 0.758). This variable is below the lower limit of 0.5 since there are elements with a small share. For example, teachers believe that digital courses (PDF, Word) do not have a strong impact on the new generations of students,

who are used to or confident in technology; they prefer videos, simulations, and VR/AR applications provided by a MOOC platform.

In terms of methods, teachers believe that online teaching should use: (a) experiential learning and teaching (conducting experiments in natural environments or at least in the virtual environment—LF 0.839); (b) investigation of the research topic—LF 0.914); (c) inquiry teaching (asking questions so that students find out the content of the course by themselves—LF 0.889) (Table 12). Other methods identified by teachers are (d) interaction in teaching and learning (training small groups that must solve mini-tasks); (e) challenge-based learning (students must find solutions to a current problem/challenge such as COVID-19), but they are not included in the model due to the multicollinearity effect. These 3 sub-items form the Methods variable, with an important influence on our model: CA—0.911).

**Table 12.** Path coefficients.

| Variable | Path Coefficients | Sample Mean | Standard Deviation | Tstatistics | $p$ Values |
|---|---|---|---|---|---|
| Methods -> Competencies | 0.897 | 0.906 | 0.114 | 7.890 | 0.000 |
| Univ -> Methods | 0.827 | 0.823 | 0.076 | 10.858 | 0.000 |
| Univ -> Competencies | 0.106 | 0.096 | 0.123 | 0.860 | 0.309 |
| Univ ->Teaching | 0.989 | 0.988 | 0.030 | 33.117 | 0.000 |

The variables identified above (Univ, Teaching, and Methods) have a significant influence on our reflective variable. Competencies variable include 5 critical elements: (a) collaboration with other students, teachers and other entities, is decisive in generating innovative ideas and appropriate ways of implementation (LF—0.863); (b) creativity—teachers are not interested in reproducing information, but in evaluating how creative the student is in solving a task and reaching a correct result (LF—0.947); (c) critical thinking—teachers have the aim to teach students how to learn, where and how to find information, how to make research and how to discover the critical aspects of a problem (LF—0.820); (d) adequate learning environment is the environment offered by IoT, MOOCs, simulations, VR/AR (LF—0.835); (e) technical skills are mandatory requirements, considering the extension of provided technologies and facilities (LF—0.863). Moreover, communication is essential in the context of globalization and multiculturality, which might offer unexpected opportunities and challenges, but it is not included in the model due to the multicollinearity effect; Competencies have a CA of 0.937.

Cronbach's Alpha (CA: Competencies—0.937, Methods—0.911, Teaching—0.758, Univ—0.886) demonstrates that our analysis is safe and consistent because each of the included questions/factors correlates with the additive result of all items (Table 1, Figure 1). The basic criterion for this operation is the value of Cronbach's Alpha index, which ranges between 0 and 1. To be considered consistent, a scale must reach a value as close to 1 as possible. The level of 0.7 is accepted as a threshold by most researchers. However. the value of Cronbach's Alpha cannot be less than 0.60. In our case, Alpha meets this condition for the 3 variables (Competencies = important elements of online teaching, Methods = online teaching methods, Univ = adaptation of educational institutions/universities to the accelerated pace of competencies in the labor market), so the factors taken into account in our analysis are relevant.

The estimation of the PLS route modeling to analyze the Univ influence on Methods and e-learning is shown in Tables 11 and 12.

Regarding the diagram, the following preliminary observations can be made:

(a)  Explanation of the variance of the target endogenous variable: The correlation coefficient ($R^2$) is 0.973 for Competencies as a latent endogenous variable. This means that the model explains 97.3% of its variance. The correlation coefficient ($R^2$) is 0.684 for Methods, meaning that the model explains 68.4% of its variance. The correlation

coefficient ($R^2$) is 0.977 for Univ, meaning that the model explains 97.7% of its variance (Table 12).

(b)  Dimensions and significance of the inner model path coefficient—the inner model is statistically significant because its path coefficient has a high value (Table 11), meaning that Univ influences Methods (0.827). Univ influences Teaching (0.989) and Methods influence e-Learning (0.897). There is an indirect effect between Univ and Competencies.

In Table 11, AVE shows how much (on average) the variables are positively correlated with their respective constructs. Therefore, to increase the AVE value, the variables showing low correlations must be excluded. If AVE > 0.5, the results of the model are consistent/appropriate. In our case, the Teaching variable is very close to the minimal limit. Composite reliability (CR) measures the fitting of PLS (partial least square), arranging the variables by their reliability. Cronbach's Alpha (CA) is similar to CR but is more sensitive to the number of variables for each construct. AVE, CR, and CA have high values for our variables (Univ, Methods, and Competencies). Taking into account the Univ, Methods and Competencies variables, our model is very consistent and confirms our hypotheses.

In Figure 2, the number of circles shows how well the variance of the latent variable is explained by the other latent variables. The numbers on the arrow are called tracking coefficients. They explain how powerful the effect of one variable is on another variable. The share of different tracking coefficients allows us to grade their relative statistical importance. From the analysis, it is observed that the adaptation of educational institutions/universities to the accelerated pace of competencies in the labor market largely determines (0.779) online teaching methods. In turn, these teaching methods have a decisive influence on the effectiveness of online learning (0.750). The methods need to be adapted to the new context (Figure 2).

### 3.2.4. Discriminant Validity

Our model is statistically powerful, as the Fornell-Larcker criterion is met (Table 13). Univ -> Teaching (0.989), Univ -> Methods (0.827), Methods -> Competencies (0.984). Another small indirect effect can be observed between Competencies and Univ (0.741).

**Table 13.** Discriminant validity.

| Variable | Fornell-Larcker Criterion | | | | Indirect Effect |
|---|---|---|---|---|---|
| | **Methods** | **Competencies** | **Teaching** | **Univ** | |
| Methods | 0.881 | | | | |
| Competencies | 0.984 | 0.867 | | | |
| Teaching | 0.890 | 0.908 | 0.654 | | |
| Univ | 0.827 | 0.848 | 0.989 | 0.843 | 0.741 |

The steps presented in Tables 13–15 allow us to assume that the indicators of the Competencies, Methods, Teaching, and Univ constructs show a strong positive correlation.

**Table 14.** Correlation between variables.

| Variable | Latent Variable Correlation | | | | R-Square | R-Square Adjusted |
|---|---|---|---|---|---|---|
| | **Methods** | **Competencies** | **Teaching** | **Univ** | | |
| Methods | 1 | | | | 0.684 | 0.681 |
| Competencies | 0.984 | 1 | | | 0.973 | 0.972 |
| Teaching | 0.890 | 0.908 | 1 | | 0.977 | 0.977 |
| Univ | 0.827 | 0.848 | 0.989 | 1 | | |

**Table 15.** Fit Summary.

| Chi-Square | |
| :---: | :---: |
| **Saturated Model** | **Estimated Model** |
| 212.426 | 219.876 |

The Chi-Square for the estimated model (219.876) is higher than the Chi-Square for the saturated model (212.426). Thus, we can state that our model fits and hypotheses H3 and H4 are accepted (Table 15).

The Variance Inflation Factor (VIF) of each construct was calculated using 5000 samples and a 95 percent bootstrapping procedure to check the significance of the variables. Figure 3 highlights an overview of the findings. The P-Values of this test are lower than 0.01. Thus, we can say that the overall VIF shows no multicollinearity between variables (Table 11). The VIF is greater than 5 for Creativity, Sustainable Univ, and Curricula, meaning that accepted multicollinearity is present. Based on the above criteria, we can state that H1, H2, H3, and H4 are accepted, with a specific mention for Univ -> Competencies, the High-Quality Univ Standards having a small influence on student Competencies variable, because the path effect is minimal, 0.106 (Table 12, Figure 3).

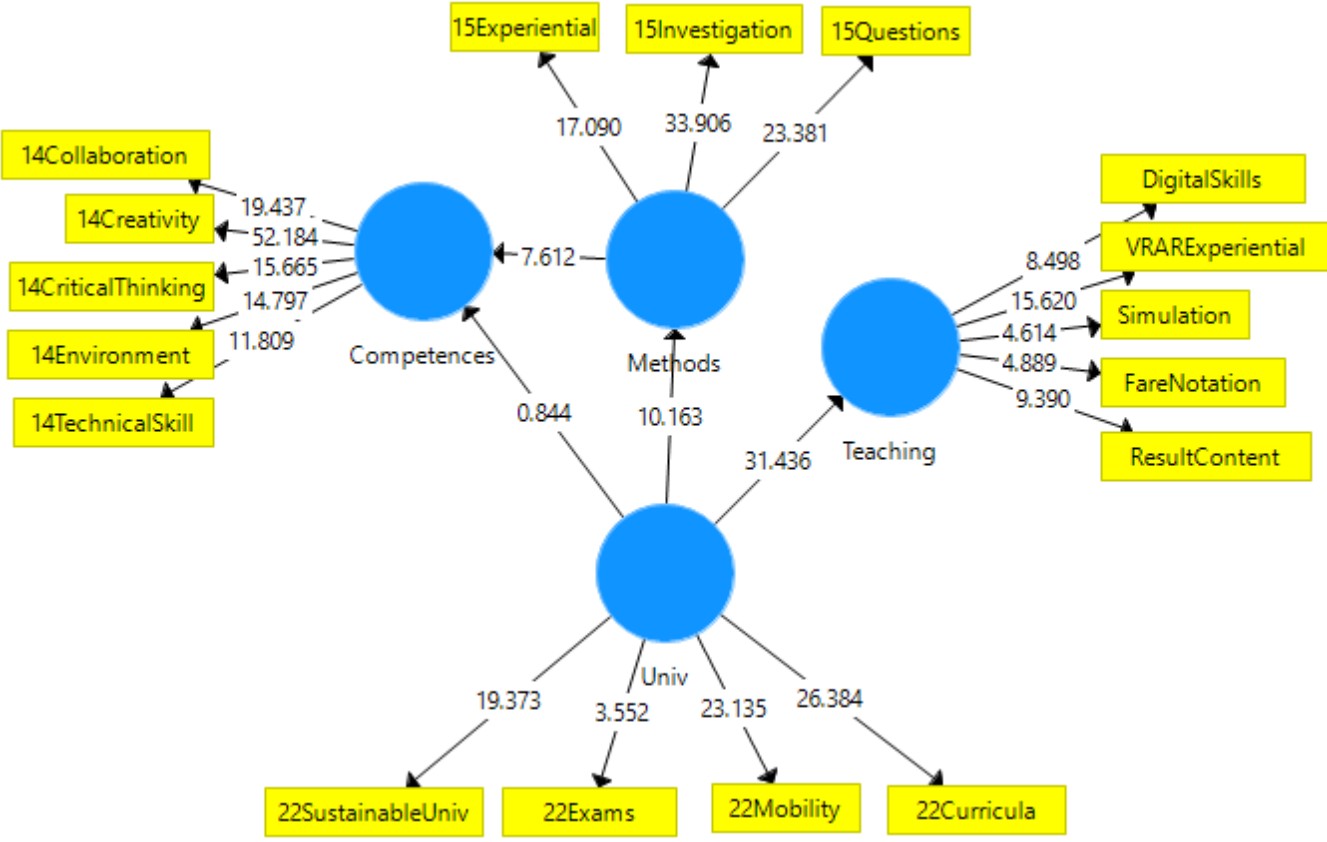

**Figure 3.** Bootstrapping significance and path coefficients.

## 4. Discussions

Based on the quantitative analysis, we can assume that online teaching was a real challenge for sports science academic teachers, confirming that the pandemic redefined the paradigm in this field too [1,2]. During the COVID-19 pandemic, most teachers faced it with good results, although they sometimes experienced loneliness, boredom, frustration, lethargy, or outrage. They accepted this situation for safety/security, thus increasing their opportunity to take additional online training, customize the organization of professional

activity, and foster decision-making autonomy. Some of them also saved time and money. All these states and attitudes give a specific sense of the internal factors conditioning the teacher's efficiency in the new conditions of the pandemic [22]. At the same time, they give new directions to university management in terms of well-being of human resource.

Teachers also experienced superficial interaction and involvement with students, as well as technical problems. Step by step, these negative issues begin to get solved due to the facilities provided by IoT and XR, which have led 45.2% of educators to say that teaching activities are the same in online and university environments. This information is in line with the results of other studies [23] but also reflects the open-mindedness of sports science teachers, who are ready to improve their teaching strategies following the challenges generated by the pandemic [48].

The main disadvantages associated with online teaching were the lack of specialized practice, lack of socio-emotional learning, lack of interactivity, health problems that determined teachers to find new solutions such as challenge-based learning to solve real problems within small groups, inquiry teaching, and experiential learning through simulations and XR. Most of the classes/courses (written, video, XR) were managed using MOOC platforms. This result confirms the extended use of IoT in sports science [51] before the pandemic and creates the prerequisites to be used both during and after the pandemic by creating a sustainable framework for the academic field.

The inferential analysis reveals that, in the current context, the university has had to become digital or integrate hybrid courses in the curricula to become entrepreneurial by establishing new partnerships with the business environment, to adapt examination methods to provide students with the transversal competencies demanded by a sustainable market, to integrate a lot of mobility programs to gain international experience and reputation based on the use of IoT in the teaching-learning process [50–53]. Sports science faced the same expectations of students about their integration in the labor market independently of the pandemic situation.

The university has to be validated by an international commitment to teaching and doing business. In this regard, new teaching methods were adopted, as can be seen in the above quantitative analysis. The academic staff aims to develop student-teacher-market collaboration, real-time open communication, students' creativity in learning and practice, and critical thinking. All these are supported by an adequate hybrid environment and technical facilities. Teachers accepted simulation software, VR, AR, and MOOC platforms only as assisting technologies in the future, but they prefer teaching in the physical environment. Online education could be only a complementary solution for HEIs and sports activities by enriching the teaching methods but cannot replace them. Our results show that the academic staff in both universities already knew and constantly used the IoT in their activity, capitalizing on developments in the field [51–53].

Our analysis highlights that the important elements of online teaching (e-learning variable) are Collaboration, Creativity, Critical Thinking, Environment, and Technical Skills (Figure 2). It also reveals that the following teaching methods are important in online teaching: (a) experiential learning and teaching (conducting experiments in natural environments or at least in the virtual environment), (b) investigation on the research topic, and (c) inquiry teaching (asking questions so that students find out the content of the course by themselves). Other influential elements are: (d) interaction in teaching and learning (training small groups that must solve mini-tasks); (e) challenge-based learning (students must find solutions to a current problem/challenge such as COVID-19). The last ones are not included in the model due to the multicollinearity effect.

Considering the types of technologies that teachers would like to use in the future and their digital skills to meet the requirements of adapting their universities to the current context, the important issues are VR and AR experiential teaching, simulation software, fair rating on e-learning platforms. Overall, teachers saw the content of the results obtained in online teaching as a crisis solution for the pandemic context. Anyway, they would prefer teaching at the academy in the classrooms and using sports facilities. Nevertheless, there

are other important elements regarding technology such as MOOCs and Digital courses (Word, PDF, etc.), which were excluded from the model due to their multicollinearity.

Our analysis shows that, in order to adapt educational institutions/universities to the accelerated pace of competencies in the labor market, the following types of activities are important:

(a)    Entrepreneurial borders targeting new competencies in organizational management (creating and developing an entrepreneurial attitude among students and improving their ability to cooperate with entrepreneurs and companies from the country and from abroad, business partnerships, clusters with the active involvement of teachers and students;

(b)    Examination type: the main focus is on gaining competencies rather than content, on transferable skills rather than the number of subjects studied, on problem-solving rather than learning by heart;

(c)    International academic achievements related to new skills and abilities adapted to the needs of the global labor market (number and outcomes of academic mobility programs) and participation of the organization in RDI programs with applications to the development of competencies) are considered priorities at the moment;

(d)    Adaptation of the curricular area to new trades and cross-cutting competencies and challenges in the labor market (inclusion of the digitization component throughout the educational and training process; distinct digitalization chapters for each study program; the number of interdisciplinary and transdisciplinary study programs.

Other important elements for high-quality university standards are:

(e)    Organization's reputation in terms of adapting the curricular area (satisfaction level of the beneficiaries, level of national and international evaluation of the organization);

(f)    Academic foresight in the field of new trades;

(g)    Access to resources for meeting the objectives of adaptation to changes (investments in online/hybrid educational platforms; investments in digital transformation (technology transfer centers, clusters, regional university consortia, digital hubs);

(h)    Student success with new competencies (number of students participating in research activities, who are enrolled as business accelerators, start-ups; graduates' entry into the labor market). They were not included in the model due to the multicollinearity effects.

Study limitations are represented by the national relevance of the results and the number of universities where the research was implemented. However, these two HEIs offered a higher number of graduates per year. The research can be continued in other faculties in the sports science area, or comparisons can be made with other specialties to open a new perspective for continuing education of the teaching staff in HEIs.

## 5. Conclusions

In the field of physical education and sports learning, the fast pace of disseminating information about human movement makes the news overwhelming. Therefore, specialists do not dispute the effectiveness of IoT, an emerging paradigm that can enhance students' independence and quality of training. Ensuring the quality of education remains one of the main coordinates of HEIs, from the perspective of sustainable education. Thus, during the pandemic, IoT technologies were discussed quite extensively because of their huge potential that can ensure a continuous increase in the quality of education in the post-pandemic period too.

**Author Contributions:** Conceptualization. R.B.-M.-Ţ., L.V., R.S. and. A.M.; methodology. R.B.-M.-Ţ. and R.S.; software. R.B.-M.-Ţ.; validation. L.V., R.S. and A.M.; formal analysis. L.V., R.S. and. A.M.; investigation. R.B.-M.-Ţ., L.V., R.S. and A.M.; resources. R.B.-M.-Ţ. and A.M.; data curation. R.B.-M.-Ţ., L.V. and R.S.; writing—original draft preparation. R.B.-M.-Ţ., L.V., R.S. and. A.M.; writing—review, and editing. R.B.-M.-Ţ., L.V., R.S. and. A.M.; visualization. A.M. and R.S.;

supervision. R.S. and A.M.; project administration. R.B.-M.-Ţ. and R.S. All authors have read and agreed to the published version of the manuscript and had an equal contribution.

**Funding:** This research received no external funding.

**Institutional Review Board Statement:** Ethical review and approval were dropped for this study since it did not involve direct intervention of the subjects.

**Informed Consent Statement:** Ethical review and approval were waived for this study since the survey was anonymous, and the respondents agreed that researchers use their answers/opinions for analysis.

**Data Availability Statement:** Not applicable.

**Conflicts of Interest:** The authors declare no conflict of interest.

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
