# Peer review of "Creating IoT-Enriched Learner-Centered Environments in Sports Science Higher Education during the Pandemic"

_sustainability, doi:10.3390/su14074339_

Round 1

Reviewer 1 Report

Dear Authors,

The manuscript is interesting; however, I am not totally convinced as it is suitable for the Sustainability journal (regarding the aim of the study, I would send it to the Education journal, for example).

Please find some suggestions to improve the manuscript (with some hints on making it closer to the journal's scope). 

In the Abstract, please specify which "Sports Science academic environment" you explore. A reader is not familiar with such a concept.
In the Introduction, the link between the aim of the study and sustainability is missing. Please add an excerpt that reveals the connection between the aim and sustainability. It could be a nice 'preface' for section 2.
In line 91, what do you mean by "sustainable higher education (HEI)"?
Could you specify the limitations of the study?
In the Conclusions, the issue of "sustainability" is also missing. 

Regarding technical aspects:
Some missing words in the text, e.g., line 13: "environment been prepared" and hard-to-read sentences appear: see lines 16-20.

The hypotheses should be numbered from 1 to 4 (not 5).

The size of Figures could be unified.

I wish you all the best in you further endavours. 

Author Response

Dear reviewer,

Thank you for your comments. We have made the corrections and we hope the manuscript meets your requirements.

Best regards,

Rares Stanescu

Reviewer 2 Report

Dear authors,

The paper looks at a topic of high interest in the recent time.

However, in order to improve the paper some major adjustments are needed, among which:

In the introduction section:

  • the term systematic review analysis is inappropriately used. You do not do a systematic literature review and you should not use this term to describe what you do.
  • the different platforms used for distance learning were NOT DEVELOPED by universities, they are USED by universities
  • revise sentence talking about X,Ya and Z generations. Are they all young?
  • You need to specify to what countries the CoHE Report refers to.

In section 2:

  • you need to define IOT, at the beginning of the section in which you start to talk about the technology (if this is a technology widely used in e-learning in sports).
  • discuss if e-learning is seen as a complementary or a replacing way to enrich teaching in sports
  • there is a lot of literature about the students reaction to technology in the e-learning process. As the paper is about teacher s perspective, more bibliographical references are needed from this perspective.

In the methodology section:

  • this section needs to be separated in Methodology and Research or re-named as Methodology and Results.
  • this section needs most of the reviewing
  • at a general level you did not present your theoretical, conceptual model. This needs to be presented (schematically), but also all the variables included need to be justified from the literature. Also the model and the relationships between variables need to be explained.
  • the universities included need to be named in the first place, the first time they appear in the text (not acronyms)
  • the questionnaire needs to be included in the appendix.
  • hypotheses are wrongly numbered
  • the first two hypotheses have no direction of influence specified (only the last two)
  • the section presenting the descriptive statistics has tot be re-written. Present the information as tables and in the text only comment and interpret it.
  • in descriptive statistics averages for each variables can be computed and rankings of variables can be presented.
  • for the inferential analysis,if you use structural equation modelling, you need to state this and you need to justify the choice of the method.
  • there is no qualitative analysis here, so please take out this statement
  • you do not need to include the same figure 3 times. Just include the figure when you present the coefficients loadings. All other aspects present them in tables (AVE, etc)
  • the discussion at page 11 bellow figure 1 is not clear. There is no obvious connection between your econometrical findings and your conclusions

Some more general comments:

  • conclusions need to include limitations of the study
  • practical and theoretical contributions need to be developed
  • one other recurring weakness in this paper is that acronyms are not explained THE FIRST TIME they appear in the text. Some are explained later in the text, others are not explained at all. This needs to be revised and acronyms to be consistently explained the first time when they appear in the text.

Author Response

Dear reviewer,

We would like to thank you for your thoughtful comments and efforts towards
improving our manuscript. We have made the corrections and we hope the manuscript will meet your requirements. 

Best regards,

Rares Stanescu

Round 2

Reviewer 1 Report

The authors made a lot of effort and followed all suggestions. I recommend the article for the publication.

Reviewer 2 Report

The authors answered to most of the reviewers' suggestions and the paper is now publishable.